# Antimicrobial V-Shaped Copper(II) Pentaiodide: Insights to Bonding Pattern and Susceptibility

**DOI:** 10.3390/molecules27196437

**Published:** 2022-09-29

**Authors:** Zehra Edis, Samir Haj Bloukh

**Affiliations:** 1Department of Pharmaceutical Sciences, College of Pharmacy and Health Science, Ajman University, Ajman P.O. Box 346, United Arab Emirates; 2Center of Medical and Bio-Allied Health Sciences Research, Ajman University, Ajman P.O. Box 346, United Arab Emirates; 3Department of Clinical Sciences, College of Pharmacy and Health Science, Ajman University, Ajman P.O. Box 346, United Arab Emirates

**Keywords:** antimicrobial agent, polyiodide, copper, antimicrobial, crown ether, pentaiodide, halogen bonding

## Abstract

Antimicrobial resistance (AMR) is a major concern for the survival of mankind. COVID-19 accelerated another silent pandemic of AMR through the uncontrolled use of antibiotics and biocides. New generations of antimicrobial agents are needed to combat resistant pathogens. Crown ethers can be used as models for drug action because they are similar to antibiotics. Iodine is a well-known microbicide but is characterized by instability and short-term effectivity. Iodine can be stabilized in the form of polyiodides that have a rich topology but are dependent on their immediate surroundings. In addition, copper has been successfully used since the beginning of history as a biocidal agent. We, therefore, combined iodine and copper with the highly selective crown ether 1,4,7,10-tetraoxacyclododecane (12-crown-4). The morphology and composition of the new pentaiodide [Cu(12-crown-4)_2_]I_5_ was investigated. Its antimicrobial activities against a selection of 10 pathogens were studied. It was found that *C. albicans* WDCM 00054 is highly susceptible to [Cu(12-crown-4)_2_]I_5_. Additionally, the compound has good to intermediate antimicrobial activity against Gram-positive and Gram-negative bacilli. The chain-like pentaiodide structure is V-shaped and consists of iodine molecules with very short covalent bonds connected to triiodides by halogen bonding. The single crystal structure is arranged across the lattice fringes in the form of ribbons or honeycombs. The susceptibility of microorganisms towards polyiodides depends on polyiodide bonding patterns with halogen-, covalent-, and non-covalent bonding.

## 1. Introduction

Antimicrobial resistance (AMR) is a serious problem faced by mankind [1]. Pathogens continuously develop mechanisms to overcome infection control measures [1,2]; it is an ongoing battle for survival between mankind and microorganisms [1]. The notorious ESKAPE pathogens and nosocomial infections are just the tip of the iceberg [1,2]. Nosocomial infections related to AMR are a serious burden on disease control morbidity and mortality globally [1,2,3]. Unfortunately, the COVID-19 pandemic accelerated AMR due to the frantic, uncontrolled use of antimicrobials and biocides [4,5,6]. In overcrowded emergency wards with overworked health personnel, the attempts to save lives ameliorated this silent AMR pandemic [5]. New generations of antimicrobials and biocides are urgently needed to maintain a good quality of life and healthcare practice. Nanotechnology seems to work well against AMR pathogens and reduce the spread of COVID-19 [7]. The use of nanomedicine with different metallic nanoparticles, such as copper, gold, and silver nanoparticles (AgNP), in particular, is popular [7,8,9,10,11,12,13,14,15]. However, AgNP synthesis remains expensive and time-consuming despite increasing investigations on facile, green methods [7,8,9,10,11,12,13,14,15]. The antimicrobial activity of AgNP depends heavily on its size, shape, and stability of the AgNP [12]. Most interestingly, resistance against AgNPs is another cause of concern [16]. Therefore, additional alternatives are needed to overcome the AMR crisis. Such alternatives must bypass the mechanisms of resistance and should exclude side effects.

As an alternative, iodine has been a known microbicide throughout the history of mankind, with various applications due to its vast availability, ease of operation, and interesting properties [17,18,19,20,21,22,23,24,25]. Iodine is used as a biocide, as iodophor, in different complex forms due to its excellent antimicrobial activities [17,18,19,20,21,22,23,24,25,26,27,28,29,30,31,32,33,34,35,36,37]. Iodine itself, and in the form of povidone iodine, has several drawbacks, such as staining, skin irritation, instability, uncontrolled iodine release, and, therefore, short-term durability in infection control [17,26,27,33]. Complexing agents ameliorate the formation of different stable polyiodides. The synthesis method determines polyiodide structures according to the formula [I_2k+n_]^n−^ (k = integer, n = 1–4) [38,39,40]. The attachment of iodine (I_2_) molecules to the stable triiodide I_3_^−^ ions (I_3_^−^) by halogen and hydrogen bonding leads to a variety of polyiodide structures [38,39,40]. Triiodides are the basic topology and are utilized in many applications in different fields [17,18,19,20,21,22,23,24,25,26,27,28,29,30,31,32,33,34,35,36,37,39,40,41,42,43,44,45]. Higher polyiodides can be synthesized depending on complexing/stabilizing agents, solvents, and available counterions [38,39,40]. Pentaiodides are one of the most interesting polyiodide classes due to their stability and varying structural features. Basic examples of pentaiodide shapes include linear, V-, L-, and Y-shaped species with [(I^−^)2(I_2_)], [(I_3_^−^)(I_2_)], and [(I_3_^−^)2(I_2_)_0.5_] units [38,39,40,45,46,47,48,49,50,51,52]. These units can be arranged in the form of isolated pentaiodide units or polymeric lines, zig-zag chains, and networks with different orders of complexity [38,40,45,46,47,48,49,50,51,52]. Polyiodide ions require good complexation agents to stabilize the resulting molecular structures. Such complexes offer scaffolds within the crystal structure and allow a rich polyiodide topology.

Macrocyclic ethers are examples of such good complexing agents and have many applications in different fields [20,21,53,54,55]. The crown ether molecules stabilize the anionic structure around the sandwiched central metal cations due to their selective binding mechanisms [20,21,52,53,54,55]. Their similarity to antibiotics makes them attractive models for drug interactions and cation transport across cell membranes [53]. Their interaction with ions influences their molecular structure; 1,4,7,10-tetraoxacyclododecane (12-crown-4), a crown ether, forms stable tri- and pentaiodides [20,21,52,53,54,55]. Different stacking orders and a number of conformations arise from their immediate surrounding in liquid and solid states [53,54,55]. According to computational investigations, 12-crown-4 self-aggregates with parallel layers of dipoles [53]. This arrangement results in weak hydrogen bonds between the C-O of one crown ether with the hydrogen atoms of a neighboring one [53]. Their structural properties ensure strong, highly selective complexation. This combination gives rise to different polyiodide topologies in accordance with previous studies [34,49,50,51,52,56,57]. 

Pentaiodides usually have better antimicrobial activities than triiodides, in accordance with our previous reports [34,49,50,51,52,56,57]. Our most recent investigation resulted in an interesting Y-shaped polymeric pentaiodide, [Cu(H_2_O)_6_(12-crown-4)_5_]I_6_ x 2I_2,_ with very good antimicrobial activities against 10 selected pathogens [50]. In this work, we were able to prepare an ambient temperature compound, [Cu(12-crown-4)_2_]I_5,_ with a higher stability and melting point with seemingly perfect crystals. The change in reaction temperature produced a different composition and polyiodide topology, verifying the interdependence of the surrounding cationic and anionic-polyiodide substructure. Our investigation confirmed that the strict control of the preparation method is relevant in polyiodide chemistry. Any deviation changes bonding patterns, polyiodide structures, and properties. This elucidates a very fluid and diverse arrangement of covalent, non-covalent, hydrogen, and halogen bonding during the attachment of iodine (I_2_) molecules to the stable triiodide I_3_^—^ions (I_3_^−^). Triiodide anions can be a mixture between “smart”, symmetric purely halogen bonded species, and asymmetric moieties, depending on the structural environment. The compound was intended to be free of hydration in expectation of better antimicrobial properties against the 10 selected pathogens. We were able to highlight the dependence between polyiodide structure and antimicrobial properties in comparison to previous investigations with [Cu(H_2_O)_6_(12-crown-4)_5_]I_6_ x 2I_2_. 

The susceptibility of microorganisms is highly dependent on molecular structure and iodine release abilities. The iodine released is ameliorated in polyiodide patterns containing more iodine units with short interatomic distances and longer connection bonds to surrounding iodide/triiodide units. The availability of porin channels in pathogens can explain the type of interaction between cell membrane constituents and polyiodide structure. Triiodide and iodide ions only move through porin channels, while iodine molecules pass through the cell membrane itself. The new compound, [Cu(12-crown-4)_2_]I_5,_ was subsequently analyzed to verify its composition and morphology. 

## 2. Results and Discussion

Polyiodides are important compounds with manifold applications. Their topology is highly dependent on cation size, solvent, complexation agents, and the structural characteristics of their surrounding molecules [38,39,40]. We used 12-crown-4 with the microbicides iodine and copper to elucidate antimicrobial properties. These are directly linked to the ability of the crown ether to selectively capture the copper ions, therefore influencing bacterial Cu homeostasis as well as Cu-detoxifying virulence factors [53,58]. In our efforts to explore the influence of crown ether-cation-polyiodide topology on microorganisms, we investigated several combinations. 12-crown-4 complex triiodides have good antimicrobial activities on pathogens due to their stability [34,56]. In this category, symmetric, halogen bonded, “smart” triiodides have the highest stability and release the iodine upon interaction with the microbial interface [34]. This behavior enhances the long-term-stability and on-site activity of antimicrobial agents based on iodine [53]. Pentaiodides and higher polyiodides are less stable, although their higher iodine content markedly increases their antimicrobial actions [50,51]. In our quest to improve the stability and action of the compounds, we chose a pentaiodide with copper ions as counterions for the crown ether. The cationic size makes copper an interesting counterion for 12-crown-4 as well. Already, small cations of lithium, zinc, and sodium have demonstrated their suitability as a counterion for 12-crown-4 in previous studies [20,21,34,53,54,55]. However, the deformation of the sandwich complexes with 12-crown-4 increases with the decreasing size and charge density of the cations [49,50,51,52,56,57]. During our investigations, we were able to structurally and biologically characterize the polymeric Cu(II)-pentaiodide [Cu(H_2_O)_6_(12-crown-4)_5_]I_6_ x 2I_2_ [50]. Our aim was to prevent any hydration around the central position. This would allow for the formation of compounds with different pentaiodide topologies and possibly enhanced antimicrobial activities. The structural characterization by X-ray methods failed due to disorders in the crystals. We tried to elucidate the polyiodide structure by TEM, SEM, STEM, UV, FTIR, Raman, and single crystal X-ray diffraction. Our primary aim was to investigate the susceptibility of 10 reference strains against the new pentaiodide [Cu(12-crown-4)_2_]I_5_ in relation to its topological features. 

### 2.1. Spectroscopical Characterization

#### 2.1.1. UV-Vis Spectroscopy

The UV-vis spectrometric analysis of the title compound [Cu(12-crown-4)_2_]I_5_ is in accordance with previous investigations (Figure 1) [33,34,35,36,37,49,50,52].

The complex [Cu(12-crown-4)_2_]I_5_ shows absorptions in the wavelength range between 200 to 500 nm (Figure 1). The spectrum is dominated by peaks of smart triiodides at 290 and 360 nm in agreement with our previous investigations (Table 1) [34,52].

The triiodide ion of the title compound is dominated by broad absorptions at 228 (s), 290 (m), and 360 (s) nm. The peak at 228 nm originates partly from the CH_2_-O-CH_2_ -groups within the 12-crown-4 molecule or triiodide ions according to previous investigations [18,19,22,52]. 

The UV-spectrum of [Rb(12-crown-4)_2_]I_5_ has, in comparison, only two peaks at 290 and 359 nm for the triiodide units (Figure 1) [49]. The peak at 360 nm is slightly red-shifted in the Cu-compound (Table 1). The smaller Cu-cation fits easier into the 12-crown-4 hole. The topology of the anionic structure is highly dependent on cationic size [49]. Overall, this bathochromic shift reveals less encapsulation, less hydrogen bonding, and less interaction between the triiodide moiety and the surroundings. Therefore, the pentaiodide unit in the title compound is probably not an isolated anion like in the Rb-complex. The crown ether sandwich in [Rb(12-crown-4)_2_]I_5_ is distorted in one direction due to the bigger size of the rubidium ion [49]. This opens up a channel within the cationic structure which allows the pentaiodide ions to fit in [49]. Therefore, the UV spectrum shows only the peaks at 290, 359, and 201 nm (Table 1) [49]. The peak at 201 nm confirms the anionic structure [(I^−^)2(I_2_)]. The peak at 201 nm belongs to iodide ions and is also available in the new Cu-complex. In both compounds, iodine is reduced to iodide ions, while Cu(I) is oxidized to Cu(II). At the same time, the size and higher charge of Cu(II) can be better complexed by 12-crown-4. The same trend is evident in previous works with 12-crown-4 [34,49,50,52,56,57]. At the same time, the fully structurally characterized Cu-hexahydrate complex is proof that Cu(II) is complex [50]. However, the title compound reveals additional peaks at 204 and 445 nm which are missing in the rubidium-pentaiodide (Table 1) [49]. The peak at 204 nm is related to iodine molecules and/or CH_2_-O-CH_2_ groups within 12-crown-4 according to previous results [52,57]. The very weak, broad peak at 445 nm originates from pentaiodide ions in agreement with preceding investigations (Table 1) [18,19,23]. This peak is missing in the rubidium pentaiodide complex because the I_5_^-^ unit is isolated and does not absorb [49]. Therefore, we can infer a chain-like anionic structure for the title compound due to its additional absorption peaks at 445 and 204 nm in contrast to the isolated pentaiodide structure (Table 1).

#### 2.1.2. Raman Spectroscopy 

The Raman analysis elucidates part of the composition of the title compound in accordance with previous studies [33,34,35,36,37,50,52]. The crown ether complex is confirmed by the Raman shift at 2926 cm^−1^ (Figure 2). 

The compound [Cu(H_2_O)_6_(12-crown-4)_5_]I_6_ x 2I_2_ revealed the Raman shift of the crown ether at 2855 cm^−1^ [50,52]. The Raman shift in the title compound appears at 2926 cm^−1^ with higher wave numbers (Figure 2). The increase in frequency indicates a decrease in chemical bond lengths within the 12-crown-4 molecule [45,46]. The chemical bonds within the crown ether complex are stronger in the title compound. The blue shift for this group, therefore, verifies compressed 12-crown-4 molecules in the title compound compared to [Cu(H_2_O)_6_(12-crown-4)_5_]I_6_ x 2I_2_ [50]. The compressive stress is due to the missing hexa-aqua complexation of the central Cu-atom, which relaxes the neighboring 12-crown-4 molecules. In the title compound, the compressive stress within the smaller lattice results in an increased wavenumber of phonons interacting with the incident photons [46]. 

The polyiodide topology manifests itself in the region between 50 to 400 cm^−1^. The Raman spectrum points to a V-shaped pentaiodide structure (Figure 2). High intensity peaks at 112 and 170 cm^−1^ indicate highly dominating symmetric stretching modes, ν_1s_, of “smart” triiodides [I-I-I^−^] and iodine (I_2_), respectively (Table 2) [22,23,28,34,39,42,45,47,50]. 

The Raman shifts of V-shaped pentaiodides are more similar to the absorption pattern of our new pentaiodide (Table 2) [39,42,45]. Linear poly-pentaiodides have no matching Raman bands for the symmetrical vibrations of the iodine units (Table 2) [22,23,28]. 

The title compound shows a shift of symmetric stretching modes of iodine towards lower wavenumbers in comparison to [Cu(H_2_O)_6_(12-crown-4)_5_]I_6_ x 2I_2_ (Table 2) [50]. The compound consisted of Y-shaped polymeric-chain pentaiodides with two iodine molecules in the form of [(I_3_^−^)2(I_2_)_0.5_] (Table 2) [50]. The shift towards slightly lower wavenumbers from 172 to 170 cm^−1^ to lower energy can manifest itself in the various topological features of the new pentaiodide. This red shift indicates a connection of the I_2_ to neighboring halogen donors [48,50] while the decrease in frequency points to a weaker bond within the I_2_ unit. The iodine molecule in the title compound is likely to have an increase in the chemical bond length, compared to the previously investigated Cu-hexaaqua-pentaiodide, within the I-I bond with a distance of around 2.78 Å (Table 2) [46,50]. This may be due to the weak interactions of the iodine molecule with neighboring iodide/triiodide ions [47]. Lastly, the red shift accompanied by narrower bands in the title compound can be an indicator of the decreased chain length of the pentaiodide units in the title compound, contrasting the polymeric pentaiodide structure within [Cu(H_2_O)_6_(12-crown-4)_5_]I_6_ x 2I_2_ [22,50]. The strong absorption at 170 cm^−1^ is also a marker for V-shaped pentaiodide ions according to the literature (Table 2) [39,42,45].

The Raman spectrum of the title compound is dominated by vibrational stretching modes of triiodide units. “Smart” triiodide ions are indicated by the strongest absorption at 112 cm^−1^ accompanied by a medium, broad shoulder at 71 cm^−1^ (Figure 2, Table 2) [34]. The latter, according to Savastano et al., is a hot band transition related to ν_2_ [43]. Another medium-sized shoulder at 144 cm^−1^ is attributed to the ν_3s_ vibrational asymmetric stretching of triiodide units within pentaiodides and is in agreement with previous reports [28,42,45,50]. This band appears due to the deviation of the triiodide ions from the “smart” D_∞h_ symmetry [34,46]. Overtones of the I_3_^−^ and the I_5_^−^ band are available as weak and very weak absorptions at 222 and 334 cm^−1^, respectively (Figure 2, Table 2) [34,43,46].

We conducted a comparative study with four crystallographically and spectroscopically characterized pentaiodide structures (Table 3).

Shestimerova et al. concluded an L-shaped structure for their polymeric pentaiodide [(3-HOC_5_H_9_NH_2_)I_5_] (Table 3) [47]. The asymmetric triiodide unit (I1-I3-I5) is bonded by weak halogen bonds through its I1 atom to an iodine molecule (I2-I4) (Table 3, Figure 3b). 

The bridging iodine molecule is characterized by short I2-I4 distances of 2.75 Å (Table 3b) [47]. The iodine and triiodide anions are both covalently bonded. At the same time, their valence vibration is shifted towards lower wavelengths at 164 and 127 cm^−1^ (Table 3) [47]. This is a result of shorter bridging distances of 3.38 to 3.40 Å between the I2-units (I2-I4) and the triiodide anions (I1-I3-I5) in comparison to 3.41 Å in the V-shaped pentaiodide [(C_10_H_16_N_2_) [I_3_]_2_·I_2_] of Reiss et al. (Table 3, Figure 3b,d) [45,47]. The triiodide units of the L-shaped pentaiodide and the V-shaped pentaiodide have angles of 177.45° and 176.43°, respectively (Table 3, Figure 3b,d) [47]. The bigger angle in the L-shaped pentaiodide, paired with the slightly shorter bridging distances, is expressed by a red shift in the Raman spectrum (Table 3) [47]. The bands 114, 127, and 164 cm^−1^ (V-shaped) moved towards shorter wavelengths of 111, 127, and 164 cm^−1^ (L-shaped, Figure 3b,d) [47]. Such a red shift indicates longer chemical bonds in the L-shaped anionic structure compared to the V-shaped pentaiodide. Indeed, the structural analysis of the L-shaped pentaiodide confirms this assumption [47]. The triiodide moieties have higher average bond lengths (2.94 Å) and the I_2_-units (2.75 Å) are slightly elongated compared to the V-shaped species (2.92 and 2.74 Å) (Table 3) [45,47]. The V-shaped I_5_^-^ polymers are halogen bonded and stacked in layers (Figure 3d) [45]. The L-shaped cis-pentaiodide chains are non-covalently bonded to the surrounding atoms (Figure 3b) [47]. The comparative study allows us to draw conclusions about our title compound based on the available Raman bands (Table 3). Our title compound is expected to consist of covalently, halogen bonded, V-shaped, and stacked pentaiodides, which are non-covalently bonded with the surrounding 12-crown-4 molecules by (O)H^…^I hydrogen bonding and (C)H^…^I van der Waals interactions (Table 3) [46,47]. Furthermore, the lack of further crystallographic data prevents the determination of the overall anionic structure in terms of verifying the exact topology.

Another V-shaped pentaiodide with isolated I_5_^−^ moieties is the compound [Rb(12-crown-4)_2_]I_5_ (Table 3, Figure 3a) [49]. The bands for the triiodide units and iodine molecules are red-shifted towards 96, 163, and 180 cm^−1^ (Table 3) [49]. Again, we can confirm from the shifts towards lower wavenumbers the longer bonds within the molecule. The iodine molecules I4-I5 (2.79 Å) and the triiodide molecule I1-I2-I3 (average 2.96 Å) are both elongated in comparison to all the compounds in Table 3 (Figure 3a) [49]. This is combined with a very short interbridge distance of 3.09 Å and an asymmetric triiodide ion with an angle of 177.30° (Table 3) [49]. We can suggest from these data that the title compound does not contain isolated pentaiodide units when the shifts are evaluated against each other. The difference arises from the larger size of the Rb^+^-cation within the 12-crown-4-sandwiched complex. The sandwich is more distorted in one direction and opens up channels for the isolated pentaiodides depicted in Figure 3a.

Comparable distances between the triiodide and iodide units are available in our Y-shaped pentaiodide [Cu(H_2_O)_6_(12-crown-4)_5_]I_6_ x 2I_2_ at 3.378 Å (Table 3, Figure 3c) [50]. These occur between the asymmetric triiodide ion I3-I4-I5 and the two I_2_-units (I1 and I2) (Figure 3c). Additionally, this compound has the longest average I_2_-bond distances (2.77 Å), paired with the shortest average interbridge distances between iodine and triiodide anions (3.38 Å) (Table 3). The resulting vibrational pattern of the molecule results in its characteristic absorptions. The Raman spectrum accordingly lacks the conclusive Raman bands at around 71, 112, 222, and 334 cm^−1^ (Table 3) [50]. Therefore, we can suggest that our pentaiodide does not contain elongated I-I bonds with short interbridge distances to asymmetric triiodide ions. The absorptions at 71 and 112 cm^−1^ in our title compound belong to “smart” triiodide ions in accordance with our previous results [34,52,56,57]. The asymmetric vibrational shoulder related to the triiodide ion at 222 cm^−1^ underlines the availability of asymmetric triiodide ions as well. In conclusion, the anionic structure consists of a mixture of a symmetrical, purely halogen bonded “smart” (I-I-I) and (I-I^…^I) asymmetric triiodide ions in accordance with previous investigations [34,52,56,57]. 

Overall, the Raman spectrum shows similarities to the absorption pattern of previously reported pentaiodides [28,39,42,45]. The above considerations indicate a V-shaped pentaiodide with mainly “smart” triiodide species (D_∞h_ symmetry) and further asymmetric I_3_^−^-units within the anionic structure of our title compound. The triiodide with D_∞h_ symmetry would be Raman inactive for the asymmetric vibration mode at 144 cm^−1^ [46]. However, the available low-intensity band appears due to slight deviations from the D_∞h_ symmetry [46]. The triiodide moieties may have average bond lengths between 2.92 to 2.93 Å (Table 3); this is verified by the blue-shifted symmetric stretching mode at 112 cm^−1^, which usually appears at 110 cm^−1^ for triiodide interbonding distances of 2.94 Å [46]. We can therefore conclude that the Raman shift at 112 cm^−1^ indicates triiodide-moieties with interbond distances of around 2.93 Å (Table 3). The bands around 222 and 334 cm^−1^ are the overtone satellites of the triiodide band at 112 cm^−1^ [46]. The iodine molecules possibly have bonds between 2.74 to 2.75 Å (Table 3). The average interbridge distances are assumed to be between 3.39 and 3.41 Å (Table 3). The pentaiodide moieties are arranged in the form of [(I_3_^−^)(I_2_)] units (Figure 2, Table 3). The chemical bond lengths within triiodide ions and 12-crown-4 units are shorter due to obvious blue shifts in an overall more compressed structure compared to the previously described [Cu(H_2_O)_6_(12-crown-4)_5_]I_6_ x 2I_2_ [22,50]. The pentaiodide units do not reveal a polymeric chain structure as observed in our previous investigation [Edis-CuI5]. The cationic structure may contain several conformers of 12-crown-4 as in [Cu(H_2_O)_6_(12-crown-4)_5_]I_6_ x 2I_2_ [50]. The availability of different 12-crown-4 conformers, as well as two types of triiodide moieties within a compressed structure, possibly resulted in disorders within the crystal and prevented crystallographic analysis.

#### 2.1.3. Fourier-Transform Infrared (FTIR) Spectroscopy 

The structure of the title compound was elucidated by FTIR analysis. The comparison with previous investigations confirmed the composition [49,50,52,59]. Table 4 highlights the absorption bands in the FTIR spectrum of pure 12-crown-4, [Cu(H_2_O)_6_(12-crown-4)_5_]I_6_ x 2I_2_ and [Rb(12-crown-4)_2_]I_5_ [49,50].

The FTIR spectrum of the title compound Cu(12-crown-4)_2_]I_5_ elucidates its structural characteristics (Table 4). Bands related to the crown ether molecule consisting of C-H, CH_2_, C-C, C-O, C-O-C, and CH-CH are available in the form of symmetric and asymmetric vibrational stretching and deformation bands in agreement with previous investigations (Table 4) [49,50,52,57]. The title compound shows higher intensity absorptions in comparison with pure 12-crown-4 ether [50]. The bigger sized, new Cu-complex has more interaction with light through the vibrational stretching modes of all the bonds. The bond lengths are increased due to complexation with Cu in the title compound. The C-H and CH_2_- modes at 2950, 2900, and 2800 cm^−1^ of pure 12-crown-4 are red-shifted compared to the Cu-complexed 12-crown-4 molecule (Table 4). The title compound reveals absorption bands at 2953, 2905, and 2859 cm^−1^ (Table 4). This absorption of such vibrational bands to higher energy points to stronger and shorter C-H bonds in the title compound. All other remaining bonds are red-shifted relative to the pure 12-crown-4 molecule (Table 4). Absorption at lower energy is manifested in longer bonds. The elongated C-O/C-O-C bonds towards the center of the sandwiched complex put a strain on the C-H and CH_2_ groups. This was also confirmed through the blue shift of the 12-crown-4 bands in the Raman spectrum of the title compound (Table 4). The asymmetric C-H stretching vibration, ν (C–H)_a_, avails another blue shift from 2950 to 2953 cm^−1^, verifying the shorter C-H bonds in the title compound compared to the pure 12-crown-4 molecule (Table 4) [50,52]. However, all other absorption bands of the title compound are red-shifted towards increased frequencies, indicating the compressive strain on all other bonds leading to the structural changes within the new sandwich complex. 

Similar bands can be found at almost the same wavelengths in the pentaiodide Rb(12-crown-4)_2_]I_5_ (Table 4) [49]. All these findings highlight the availability of a sandwiched Cu(12-crown-4)_2_]I_5_ -complex. There is a red-shifted vibrational stretching mode of the CH-CH groups at 841 cm^−1^ compared to [Cu(H_2_O)_6_(12-crown-4)_5_]I_6_ x 2I_2_ (844 cm^−1^), pure 12-crown-4 (848 cm^−1^), and Rb(12-crown-4)_2_]I_5_ (849 cm^−1^) (Table 4) [49,50]. Additional red shift trends are observed for C-H symmetric stretching modes towards the title compound at 2859 cm^−1^ in analogy to Rb(12-crown-4)_2_]I_5_ (2865 cm^−1^) and [Cu(H_2_O)_6_(12-crown-4)_5_]I_6_ x 2I_2_ (2863 cm^−1^) (Table 4) [49,50]. The C-O vibrational bands of the other compounds in Table 4 appear blue-shifted in comparison to the title compound. Pure 12-crown-4, Rb(12-crown-4)_2_]I_5_, [Cu(H_2_O)_6_(12-crown-4)_5_]I_6_ x 2I_2,_ and the title compound show these modes at 1100, 1095, 1093, and 1090 cm^−1^, respectively (Table 4) [49,50]. The title compound has, therefore, remarkably longer C-O and CH-CH-bonds than the other compounds, which probably result from the increased strain on the C-O bonds within the crown ether molecules. The data suggests highly complexed 12-crown-4 molecules surrounding the Cu-ion (Table 4) [50]. The bands at 841 and 548 cm^−1^ are due to stretching vibrations ν56 and ν57/ν16 originating from the crown ether molecules [50,55].

The spectrum of [Cu(H_2_O)_6_(12-crown-4)_5_]I_6_ x 2I_2_ contains broad, high-intensity absorption signals of water molecules, which are incorporated into the structure (Table 4) [50]. Hydration within the molecule [Cu(H_2_O)_6_(12-crown-4)_5_]I_6_ x 2I_2_ was verified by the broad absorption band at 3749 and 3418 cm^−1^ (Table 4) [50]. The two latter bands, and a band at 628 cm^−1^, are absent in the title compound. The weak FTIR absorptions at 3393, 3240, and 3179 cm^−1^ are due to the stretching vibrations of hydroxyl groups from possibly entrapped methanol molecules during the crystallization process or point to weak hydrogen bonding between 12-crown-4 molecules [53]. The absorption at 1638 cm^−1^ appears due to vibrations of the C-H or O^…..^H. The bands above 3400 cm^−1^ are missing in the spectrum of the title compound. This confirms the absence of hydration within the crystal structure, unlike in [Cu(H_2_O)_6_(12-crown-4)_5_]I_6_ · 2I_2_. Disorder within the structure due to the availability of different 12-crown-4-conformations impedes the crystal structure analysis of our title compound [20].

#### 2.1.4. X-ray Diffraction (XRD)

The XRD spectrum of the title compound is characterized by sharp diffraction peaks related to iodine (Figure 4). 

The strongest peaks, 2θ *=* 15.3° (111) and 25.3° (220), in the XRD analysis belong to iodine and pentaiodide species, respectively (Figure 3) [25,29,30,33]. Further peaks at 17.1 (301), 47.4, 39.1, 31 (222), and 23.1° are assigned to iodine molecules according to previous investigations (Table 5) [24,25,29,30,33]. 

The XRD diffraction pattern with sharp confine peaks confirms the availability of crystalline phases and verifies the purity of our title compound (Figure 4, Table 5). There are similarities to the CuI diffraction data but all peaks appear at different ranges (JCPDS Card 06-0246) [24]. The authors reported main peaks at 25.43, 29.45, 42.15, and 49.8° [24]. Our XRD analysis shows a shift towards smaller 2Theta values (Table 5). The shift to lower diffraction values implies expansion of the lattice in comparison to pure CuI [24]. The authors also mention impurities at 52.27, 61.16, and 69.34° [24]. Our compound has no peaks resulting from impurities, which arise due to the reaction of CuI with air or moisture [24]. Therefore, we can affirm that our compound is free of oxidation products and impurities, as well as unreacted CuI. 

### 2.2. Elemental Composition and Morphological Examination

#### 2.2.1. Scanning Electron Microscope (SEM)

SEM analysis was employed to investigate the morphology of the title compound (Figure 4).

[Cu(12-crown-4)_2_]I_5_ reveals a highly crystalline pattern in the SEM analysis (Figure 5). There are stacked, rectangular to almost cubic layers, above each other which have smooth surfaces.

#### 2.2.2. Transmission Electron Microscope (TEM) and Scanning Transmission Electron Microscope (STEM) Analysis with Elemental Mapping

TEM, STEM, and Energy-Dispersive X-ray Spectroscopic (EDS) analyses were carried out to elucidate the composition and morphology of our title compound (Figure 6).

Figure 6a reveals a highly single-crystalline morphology and includes some stacking faults. These patterns can be clearly seen in Figure 6b. The main layer is a wave-like structure with parallel lattice fringes along {001}. The average d-spacing is 0.274 nm (2.74 Å). Figure 6b depicts another wave- or barrel-like enhanced area, which seems to be further layers on the main crystal lattice or stacking faults. The barrel-like structure is arranged at an angle of around 115° to the main lattice fringe. 

The EDS analysis verifies the purity of our compound (Figure 6d). Iodine, copper, carbon, and oxygen appear in the EDS in decreasing order. Cu is increased due to the use of a copper sample holder. The percentage of oxygen, in comparison to the compound [Cu(H_2_O)_6_(12-crown-4)_5_]I_6_ x 2I_2,_ verifies the absence of hydration in the structure [50]. All the related atoms are distributed evenly in the EDS layered images (Figure 6e,f). The iodine atom distribution in Figure 6d shows that the polyiodide structure is arranged around the surroundings of the 12-crown-4-complexed Cu(II)-ions throughout the crystal.

The barrel-like wave structure in Figure 7 reveals an interesting, inter-woven structure.

Due to the strain in the enhanced structure on the top of the darker waves, rows of white strings, interwoven and shifted against each other, are clearly visible in the lower part of Figure 7. The strings seem to actually be two bigger structures in the extremities that are connected with each other by thin lines. The strings have an average length of 0.45 nm (5 Å) and are an average length of 0.35 nm (3.5 Å) away from each other. Each barrel-like wave has an average diameter of 1.15 nm. Six strings are stacked above each other in a shifted line; these lines have an average total length of 1.8 nm in each barrel from up to down and are arranged at an average angle of 179.8°. Around five such lines make up a barrel-like structure, shifted against each other from left to right. A total of six lines arrange themselves into one barrel-like structure at an angle of 115°.

The “basic” crystal lattice is neatly ordered along the lattice fringes as manifested in the 1274 % magnification of Figure 6c, depicted here as Figure 8a.

The structure contains at least two different crystal structure arrangements (Figure 8). These resemble either ribbons or honeycomb packing with hexagonal shapes along the lattice fringes (Figure 8a). The lines, which were present as strings in Figure 7, now look like white rectangles (Figure 8a). Two white rectangles are connected with each other in a “honeycomb” from up to down through an average diameter of 0.4 nm (4 Å). The white rectangles are arranged in the form of ribbons running from up-left to bottom-right. A closer look reveals that the white rectangles on the ribbons are running down in a staircase manner. Each rectangle is connected to another rectangle on the opposite side, which is part of the ribbon but in the opposite direction (Figure 8c). 

The center of the honeycomb is supposedly filled with a copper ion, which is seen on the spot pattern in Figure 8b. It is purportedly surrounded by two 12-crown-4 molecules which are seen as rectangles/strings on each side in the form of a sandwich (Figure 8c). These 12-crown-4 molecules are part of the ribbon on each end. The copper ion is connected with the oxygen atoms of the complexing ether. The two 12-crown-4 molecules are shifted towards each other within a sandwich (Figure 8c). One of them is always up while the other is below the plane (Figure 8). The lower sandwich complex in the ribbon is arranged in the same manner, but the 12-crown-4 are in opposite arrangement towards the plane (Figure 8c). The anionic structure arranges itself in the cavities, or channels, around the cationic structure.

A comparison of the distances in the electron diffraction analysis reveals interesting data. The electron diffraction shows a highly single-crystalline, primitive unit cell pattern (Figure 9).

There are three bright spots (named here as 1,2, and 3) in the center of the electron diffraction pattern (Figure 9). These spots are arranged perpendicular to the lattice fringe. The brightest central spot (2, {000}) has two slightly different distances to the two neighboring points 1 and 3 (Figure 9). The lower spot (1) and the upper spot (3) are connected with an average angle of 179.99° (1-2-3) (Figure 9). The distances are slightly different with 4.4731 1/nm and 4.66 1/nm for 1-2 and 2-3, respectively (Figure 8). Point 1 has a distance of 4.25 1/nm to the neighboring unnumbered weak spot 1‘ towards the lower left side (Figure 9). 

The bright spots 1, 2, and 3 are arranged within an average angle of 179.97°, with weaker points above (6, 7) and below (4, 5) along the lattice fringe (Figure 9). The weaker spots 6-7 and 4-5 are parallel lines to 1-2-3. The upper, weakly-appearing parallel line (6-7) is away from the central line 1-2-3 around an average of 6.95 1/nm along the lattice fringe (Figure 9). The lower-right arranged line (4-5) is on average 7.3 1/nm away from the central arrangement. The distances between 2 and points 4, 5, 6, and 7 are 7.18, 7.31, 6.92, and 7.15 1/nm, respectively (Figure 8). Distances between the points 4′-4, 4-5, 5-5′, and 6-7 are 4.302, 4.692, 4.67, and 4.61 1/nm, respectively (Figure 9). There are also several weak spots, which seem to be either disorders or further structural determinants in the structure. Figure 9 is along the {001} zone axis. The points 1, 6, and 7 are related to (−200), (−1−30), and (1−30).

In conclusion, the electron diffraction pattern shows parallel planes of points along the lattice fringes (7-3-5‘ and 6-2-5) (Figure 9). There are slightly different distances between the atoms within these planes with an overall average of around 4.46 1/nm. The atoms along the lattice fringes display an elongated average of 7.135 1/nm. The atoms along the lattice fringes (7-3-5‘ and 6-2-5) are farther away from each other than the atoms arranged perpendicular (5-6, 1-2-3, and 4-5-5‘) to the lattice (Figure 9). All atoms in these lines are lined up at angles of 180° (Figure 9).

### 2.3. Antimicrobial Testing

An Agar Well and disc diffusion assay against 10 reference microorganisms was utilized to analyze the antimicrobial properties of the title compound. Gram-positive bacteria included the reference strains *S. pneumonia* ATCC 49619, *S. aureus* ATCC 25923, *S. pyogenes* ATCC 19615, *E. faecalis* ATCC 29212, and *B. subtilis* WDCM0003. Gram-negative bacteria consisted of *E. coli* WDCM 00013 Vitroids, *P. mirabilis* ATCC 29906, *P. aeruginosa* WDCM 00026 Vitroids, and *K. pneumonia* WDCM00097 Vitroids. The antifungal activities of the title compound were investigated against *C. albicans* WDCM 00054 Vitroids. The results were compared to common antibiotics cefotaxime, amikacin, streptomycin, gentamicin, and nystatin, which were utilized as positive controls. Acetonitrile and methanol were used as negative controls. They showed no zone of inhibition (ZOI) and are not included in Table 6.

The title compound in AW studies shows good results in comparison to the positive control antibiotics (Table 6). The Gram-negative pathogens are less susceptible than the Gram-positive microorganisms. The highest antimicrobial action was exerted against the fungal reference strain *C. albicans* WDCM 00054 with a ZOI of 45 mm (Table 6, Figure 10). 

The results are similar to the antimicrobial action of [Cu(12-crown-4)_5_(H_2_O)_6_]I_6_ x 2I_2_ in AW studies (Table 6) [50]. *S. pyogenes* ATCC 19615 was more susceptible against the title compound than the hexa-aqua complex with ZOIs of 25 and 20 mm, respectively (Table 6). The disc diffusion assays were done with concentrations of 11, 5.5, and 2.75 µg/mL (Table 6, Figure 11). 

Disc diffusion (DD) studies with the title compound reveal the highest antimicrobial action with a ZOI of 25 mm against the fungus *C. albicans* WDCM 00054 and the Gram-positive *S. pneumoniae* ATCC 49619, followed by Gram-negative *P. mirabilis* ATCC 29906 (ZOI = 20 mm), and *B. subtilis* WDCM 00003 (ZOI = 18 mm) (Table 6, Figure 11). *K. pneumoniae* WDCM 00097 and *E. faecalis* ATCC 29212 are similarly susceptible with a ZOI = 16 mm, closely followed by the Gram-negative *E. coli* WDCM 00013 with a ZOI = 15 mm, and finally, *S. pneumoniae* ATCC 49619 (ZOI = 18) (Table 6). The microbicidal action decreases in the order *S. pyogenes* ATCC 19615 (ZOI = 13), *P. aeruginosa* WDCM 00026 (ZOI = 12), and *S. aureus* ATTC 25932 (ZOI = 13) (Table 6). *S. pneumoniae* ATCC 49619 and *P. mirabilis* ATCC 29906 show higher susceptibility in DD studies than in AW (Table 6). They are also more susceptible to the title compound compared to [Cu(12-crown-4)_5_(H_2_O)_6_]I_6_ x 2I_2_ [50]. *P. aeruginosa* WDCM 00026 (ZOI = 12) is similarly resistant to both compounds. The title compound has remarkably lower antifungal properties in contrast to the hexa-aqua-pentaiodide (ZOI 25 mm/53 mm) (Table 6).

The susceptibility of microorganisms towards our title compound reveals no clear trends. The general pattern of antimicrobial activity appears when the results of all concentrations of 11, 5.5, and 2.75 µg/mL are included (Table 6). In AW studies, the microbial susceptibility against our compound can be ordered as fungi (*C. albicans* WDCM 00054), Gram-positive cocci (*S. pyogenes* ATCC 19615, *S. pneumoniae* ATCC 49619, *E. faecalis* ATCC 29212, and *S. aureus* ATTC 25932), and lipophilic bacilli (*B. subtilis* WDCM 00003) in descending order (Table 6). These are followed by Gram-negative bacilli (*P. aeruginosa* WDCM 00026, *E. coli* WDCM 00013, and *P. mirabilis* ATCC 29906) (Table 6). A closer look matches an increasing microbial size, decreasing motility by fewer number of flagellae, hydrophilicity, and availability of porin channels in the order *P. mirabilis* ATCC 29906 towards *P. aeruginosa* WDCM 00026. The DD method reveals the highest antimicrobial action against fungi, then Gram-positive, lipophilic bacilli (*B. subtilis* WDCM 00003) without porin channels. These are followed by Gram-negative bacilli *K. pneumoniae* WDCM 00097 (16/8 mm), *E. coli* WDCM 00013 (15/9 mm), and *P. mirabilis* ATCC 29906 (20/0 mm) (Table 6). Again, the susceptibility is enhanced in bigger sized bacterial rods with less motility. *K. pneumoniae* WDCM 00097 (16/8 mm) and *E. coli* WDCM 00013 are hydrophilic and also have porin channels (Table 6). In conclusion, iodide, triiodide, 12-crown-4, and Cu(12-crown-4)-ions could be part of the killing action in DD studies. The lowest susceptibility is seen in Gram-positive streptococci *S. pneumoniae* ATCC 49619, *E. faecalis* ATCC 29212, *S. pyogenes* ATCC 19615, and finally, the staphylococci *S. aureus* ATTC 25932 (Table 6).

The trends in DD, and especially AW, show less importance of porin channels in the title compound. Such water-filled channels in the bacterial membrane allow for diffusion of the small, hydrophilic moieties iodide, triiodide, 12-crown-4, Cu-ions, and Cu(12-crown-4)-ions into the pathogens [58]. We can thus conclude that the hydrophilic ions are not important in the killing action in AW and are potentially relevant in DD studies in the title compound. Instead, iodine molecules are freed by the deformation of the complex molecule through dipole-dipole and electrostatic interactions. These happen between the positively polarized carbon atoms in the 12-crown-4 molecules and the oxygen atoms in the negatively charged microbial membranes [34]. These are available in the negatively charged peptidoglycan layers of the Gram-positive bacteria. Then, iodine molecules move by passive diffusion through the membranes directly into the cytoplasm. They exert their antimicrobial action by protein oxidation, efflux pump deactivation, and iodination of cell membrane fatty acids [1,50]. The antimicrobial activity in [Cu(12-crown-4)_5_(H_2_O)_6_]I_6_ x 2I_2_ is higher in almost all cases due to the higher flexibility, bigger size, and hydration of the molecule [50].

The selection of microorganisms shows less susceptibility towards the title compound in comparison to [Cu(H_2_O)_6_(12-crown-4)_5_]I_6_ x 2I_2_ [50]. The reasons may be the lack of hydration and the shorter bond lengths due to the higher compressive stress within the title compound. The tighter, strongly coordinated, smaller molecule without hydration and shorter bonds is less interactive with the microbial surfaces. Again, the decreased antimicrobial properties of the title compound in contrast to [Cu(H_2_O)_6_(12-crown-4)_5_]I_6_ x 2I_2_ verifies the absence of hydration in the structure [50]. The water molecules can interact with the molecules on the microbial cell membranes by hydrogen bonding or electrostatic interactions. This results in a distortion of the complex and, finally, the release of iodine and Cu-ions. The polyiodides in the title compound are more complex and cannot be released as in the bigger, hydrated molecule [Cu(H_2_O)_6_(12-crown-4)_5_]I_6_ x 2I_2_ [50]. This was also verified by a blue shift of triiodide-moieties in the title compound in comparison to the hexaaqua-copper compound. The Y-shaped pentaiodide with elongated bonds between the central iodide ion and the three iodine molecules releases iodine molecules more easily by any deformation. The title compound has, in conclusion, no Y-shape and shorter bonds within the V-shaped pentaiodide moieties as verified by Raman and further comparative considerations (Table 3). The release of Cu-ions into the cell membrane is another intriguing factor to discuss. 

Cu ions are transported by porin channels in Gram-negative bacteria and cause cell death [58]. The Gram-negative bacilli *K. pneumoniae* WDCM 00097 and *E. coli* WDCM 00013 are more susceptible to [Cu(H_2_O)_6_(12-crown-4)_5_]I_6_ x 2I_2_ in comparison to [Cu(12-crown-4)_2_]I_5_ itself (Table 6). The Cu-ions in the larger hexaaqua-complex are easier released than in the smaller title compound. The released Cu-ions pass through the porin channels and result in higher antibacterial action. The title complex features a stronger encapsulation of Cu-ions, which will release fewer copper ions if any. The bacterial Cu-homeostasis and the Cu-detoxifying virulence factors, especially in the Gram-negative *K. pneumoniae* WDCM 00097 and *E. coli* WDCM 00013, prevail, resulting in less susceptibility towards [Cu(12-crown-4)_2_]I_5_.

## 3. Materials and Methods

### 3.1. Materials

Copper iodide (CuI), iodine (≥99.0%), and Mueller Hinton Broth (MHB) were purchased from Sigma Aldrich (Gillingham, UK). Disposable sterilized Petri dishes with Mueller Hinton II agar and McFarland standard sets were received from Liofilchem Diagnostici (Roseto degli Abruzzi (TE), Italy). 1,4,7,10-Tetraoxacyclododecan (12-crown-4) was obtained from Sigma-Aldrich Chemical Co. (St. Louis, MO, USA). The bacterial strains *E. coli* WDCM 00013 Vitroids, *P. aeruginosa* WDCM 00026 Vitroids, *K. pneumoniae* WDCM00097 Vitroids, *C. albicans* WDCM 00054 Vitroids, and *Bacillus subtilis* WDCM0003 Vitroids were purchased from Sigma-Aldrich Chemical Co. (St. Louis, MO, USA). *S. pneumoniae* ATCC 49619, *S. aureus* ATCC 25923, *E. faecalis* ATCC 29212, *S. pyogenes* ATCC 19615, and *P. mirabilis* ATCC 29906 were bought from Liofilchem (Roseto degli Abruzzi (TE), Italy). Gentamicin (9125, 30 µg/disc), cefotaxime (9017, 30 µg/disc), chloramphenicol (9128, 10 µg/disc), streptomycin (SD031, 10 µg/disc), amikacin (30 µg/disc), and nystatin (9078, 100 IU/disc) were received from Liofilchem (Roseto degli Abruzzi (TE), Italy). Sterile filter paper discs (diameter of 6 mm) were purchased from Himedia, India. Methanol was bought from EMSURE (Merck KGaA, Darmstadt, Germany) and Acetonitrile was obtained from MTEDIA (TEDIA Company, Fairfield, Ohio, USA). All reagents were of analytical grade and utilized as received. Ultrapure water was used.

### 3.2. Synthesis of [Cu(12-crown-4)_2_]I_5_

In total, 0.12 g (0.63 mmol) of CuI was dissolved in 10 mL acetonitrile by stirring at room temperature for 15 min while covered with parafilm. Meanwhile, 0.32 g (1.26 mmol) I_2_ was dissolved in 20 mL methanol with stirring at room temperature and covered with parafilm. The two solutions were mixed under continuous stirring at room temperature. Into this solution, 0.2 mL (1.26 mmol) 12-crown-4 was added dropwise within 30 min. Stirring continued until the solution was clear and everything was dissolved. Dark brownish-red crystals appeared after 2-3 days at room temperature through slow evaporation. The crystals resulted in a percentage yield of 85%.

### 3.3. Characterization of [Cu(12-crown-4)_2_]I_5_

The complex underwent microstructural analysis by TEM and SEM. STEM-EDS was used to confirm the purity of the compound. XRD, Raman, UV-Vis, and FTIR were utilized for analyzing the composition and functional groups. The analytical methods confirmed the composition and purity of our compound [Cu(12-crown-4)_2_]I_5_. The complex was stored in the fridge for 18 months in a closed glass bottle without changing its composition and morphology.

#### 3.3.1. UV-Vis Spectrophotometry (UV-Vis)

The UV-Vis analysis of [Cu(12-crown-4)_2_]I_5_ was performed with a UV-Vis spectrophotometer, model 2600i, from Shimadzu (Kyoto, Japan) at a wavelength ranging from 200 to 800 nm.

#### 3.3.2. Raman Spectroscopy 

The Raman analysis was done at RT on RENISHAW (Gloucestershire, UK) with an optical microscope. The sample was used to fill a 1 cm × 1 cm sized cuvette and placed into the laser beam. The excitation of the solid-state laser beam was 514 nm and operated with a 50× objective of a confocal microscope and a 2micron spot diameter. The CCD-based monochromator with a spectral range of 50–3400 cm^−1^ collected the scattered light. The output power was 15%, the integration time was -300s and the spectral resolution was −1 cm^−1^.

#### 3.3.3. Fourier-Transform Infrared (FTIR) Spectroscopy 

The FTIR analysis of the title compound was performed on an ATR IR spectrometer with a diamond window in the range of 400 to 4000 cm^−1^ (Shimadzu, Kyoto, Japan). 

#### 3.3.4. X-ray Diffraction (XRD)

The X-ray diffraction analysis of the title compound was carried out on an XRD from BRUKER (D8 Advance, Karlsruhe, Germany). The measurements were executed by Cu radiation with a wavelength of 1.54060, coupled with Two Theta/Theta, a step size of 0.03, and a time/step of 0.5 s.

#### 3.3.5. Scanning Electron Microscopy (SEM) and Energy-Dispersive X-ray Spectroscopy (EDS)

The title compound was analyzed by scanning electron microscopy (SEM), using model JEOL JSM-IT200 from JEOL ASIA PTE. Ltd. (Singapore) at 20 kV without coating. A drop of [Cu(12-crown-4)_2_]I_5_ dispersed in distilled water was dropped on a carbon-coated copper grid and dried under ambient conditions. The SEM analysis revealed the morphology of the sample. 

#### 3.3.6. Transmission Electron Microscopy (TEM) and Energy-Dispersive X-ray Spectroscopy (STEM-EDS)

The title compound was analyzed by scanning electron microscopy (SEM), using model JEOL JEM-2100F from JEOL ASIA PTE. Ltd. (Singapore) with a ZrO/W(100) Schottky Type emitter and an acceleration voltage of 200 kV. A drop of [Cu(12-crown-4)_2_]I_5_ dispersed in distilled water was dropped on a carbon-coated copper grid and dried under ambient conditions. The TEM delivered information related to the structure and morphology of [Cu(12-crown-4)_2_]I_5_. The STEM-EDS analysis confirmed the purity and composition of the sample. 

### 3.4. Bacterial Strains and Culturing

The reference microbial strains of *S. pneumoniae* ATCC 49619, *S. aureus* ATCC 25923, *E. faecalis* ATCC 29212, *S. pyogenes* ATCC 19615, *Bacillus subtilis* WDCM 0003 Vitroids, *P. mirabilis* ATCC 29906, *E. coli* WDCM 00013 Vitroids, *P. aeruginosa* WDCM 00026 Vitroids, *K. pneumoniae* WDCM 00097 Vitroids, and *C. albicans* WDCM 00054 Vitroids were utilized for the antimicrobial testing of the title compound. The reference strains were stored at −20 °C. The inoculation was done by adding the fresh microbes into MHB and keeping these suspensions at 4 °C until further use.

### 3.5. Antimicrobial Testing

The susceptibility of nine reference bacterial strains (*S. pneumoniae* ATCC 49619, *S. aureus* ATCC 25923, *S. pyogenes ATCC* 19615, *E. faecalis* ATCC 29212, and *B. subtilis* WDCM 00003, *P. mirabilis* ATCC 29906, *P. aeruginosa* WDCM 00026, *E. coli* WDCM 00013, and *K. pneumoniae* WDCM 00097) against the title compound was compared to the antibiotic gentamicin as a positive control. *C. albicans* WDCM 00054 was utilized to investigate the antifungal activity of the title compound in comparison to the antibiotic nystatin as a positive control. The negative controls methanol and acetonitrile showed zero inhibition zones and were not mentioned further. The antimicrobial tests on discs were repeated three times and the average of these independent experiments was used in this investigation.

#### 3.5.1. Procedure for Zone of Inhibition Plate Studies

The title compound was used against the selected pathogens according to the zone of inhibition plate method [59]. The selected bacterial strains were suspended in 10 mL MHB and incubated for 2 to 4 h at 37 °C. The fungal strain *C. albicans* WDCM 00054 was incubated on Sabouraud Dextrose broth at 30 °C. All microbial cultures were adjusted to a 0.5 McFarland standard. Disposable, sterile Petri dishes with MHA were uniformly seeded with 100 μL microbial culture by using sterile cotton swabs. *C. albicans* WDCM 00054 was incubated on agar plates at 30 °C for 24 h. The prepared dishes were then dried for 10 min and subsequently used for antimicrobial testing.

#### 3.5.2. Disc Diffusion Method

The antimicrobial testing was done according to the recommendations of the Clinical and Laboratory Standards Institute (CLSI) [60]. Antibiotic discs of gentamycin and nystatin were utilized to compare the results. Sterile filter paper discs (Himedia, India) with a diameter of 6 mm were coated for 24 h with 2 mL of the title compound of known concentrations (50 µg/mL (1), 2 mL of 25 µg/mL (2), and 2 mL of 12.5 µg/mL (3)). Afterward, the discs were dried for 24 h under ambient conditions. A ruler was utilized to measure the diameter of the zone of inhibition (ZOI) to the nearest millimeter. The susceptibility of the microbial reference strains against the title compound can be elucidated from the ZOI around the disc. The reference strains are resistant if there is no inhibition zone and the ZOI equals zero. 

### 3.6. Statistical Analysis

SPSS software (version 17.0, SPSS Inc., Chicago, IL, USA) was employed in our statistical analysis. The data are represented as mean values. The statistical significance between groups was calculated by one-way ANOVA. Any value of *p* < 0.05 was appraised as statistically significant.

## 4. Conclusions

Polyiodides combined with complexing agents and copper ions have good antimicrobial activities and can provide future resistance against AMR. We used a crown ether due to its ability to act as a model for antibiotics and their drug actions. Antibiotics influence cation transport mechanisms across membranes due to their selective ion capture, which also has an impact on detoxifying mechanisms based inside the bacterial cell. The selectivity and strong complexing ability of 12-crown-4 allow for the controlled synthesis of antimicrobial compounds based on different polyiodide topologies. The crown ether strongly encapsulates copper ions and influences Cu-homeostasis and Cu-detoxifying mechanisms by the bacterial cell. We prepared the new compound [Cu(12-crown-4)_2_]I_5_ under slightly different reaction conditions and compared it to our previous pentaiodide compound [Cu(H_2_O)_6_(12-crown-4)_5_]I_6_ x 2I_2_. The analytical results are surprising, revealing a completely different morphology, topology, and antimicrobial properties. The new compound contains a highly crystalline structure with different patterns in the form of honeycombs and ladders. The availability of halogen-bonded species with even ”smart”, pure halogen-bonded triiodides influence the antimicrobial actions. Polyiodide topologies with mainly iodine molecules and a Y-shape, such as in our previous compound, [Cu(H_2_O)_6_(12-crown-4)_5_]I_6_ x 2I_2;,_ have a supposedly higher release of iodine. This results in the higher susceptibility of *C. albicans* WDCM 00054 and Gram-positive pathogens due to their cell membrane structure. The bigger size of the molecule and the availability of hydration increased the interaction with the cell membrane molecules. At the same time, Cu-ions seem to be more easily released, to move through the porin channels in Gram-negative bacteria, and cause higher inhibition of the Gram-negative bacilli *K. pneumoniae* WDCM 00097 and *E. coli* WDCM 00013. Our new complex is a smaller molecule with shorter bonds throughout and is free of hydration. The shorter, halogen-bonded V-shaped pentaiodide units are not readily released. The crown ether also strongly encapsulates the copper-ions and prevents their movement through porin channels. In general, *C. albicans* WDCM 00054 and rod-shaped bacilli were more susceptible to [Cu(12-crown-4)_2_]I_5_ than Gram-positive cocci. The synthesis of polyiodide compounds, especially for susceptibility assays and antimicrobial uses, is very fluid and requires strict control of the reaction conditions. Iodine, triiodide-, and iodide- release depend on the bonding pattern, polyiodide shape, surrounding structure, and structural characteristics of the compound. Future investigations need to implement such considerations for antimicrobial uses.

## Figures and Tables

**Figure 1 molecules-27-06437-f001:**
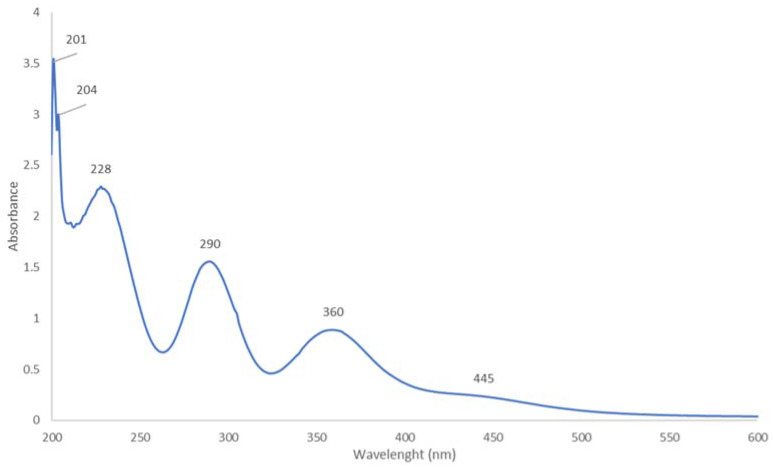
The UV-vis analysis of [Cu(12-crown-4)_2_]I_5_.

**Figure 2 molecules-27-06437-f002:**
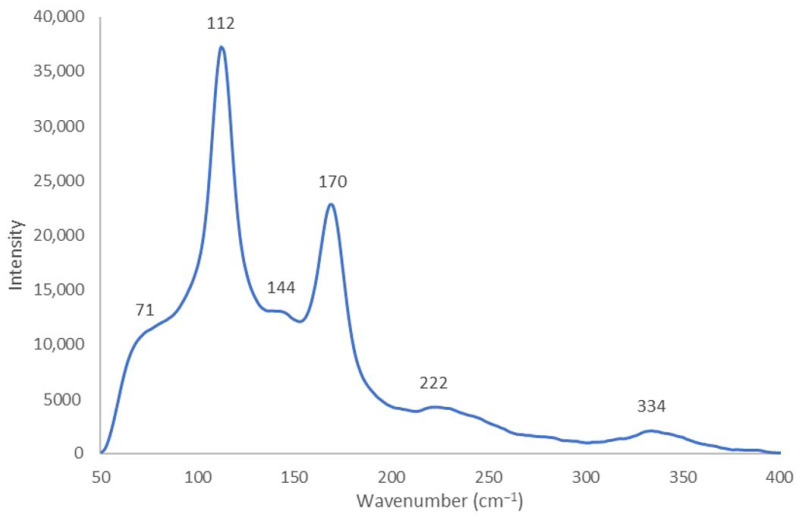
The Raman spectroscopic analysis of [Cu(12-crown-4)_2_]I_5_.

**Figure 3 molecules-27-06437-f003:**
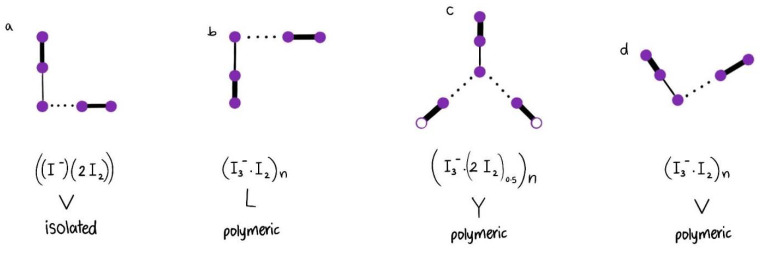
Comparative analysis of pentaiodides. (**a**) [Rb(12-crown-4)_2_]I_5_ [49]; (**b**) [(3-HOC_5_H_9_NH_2_)I_5_] [47]; (**c**) [Cu(H_2_O)_6_(12-crown-4)_5_]I_6_ x 2I_2_ [50]; (**d**) [(C_10_H_16_N_2_) [I_3_]_2_·I_2_] [45].

**Figure 4 molecules-27-06437-f004:**
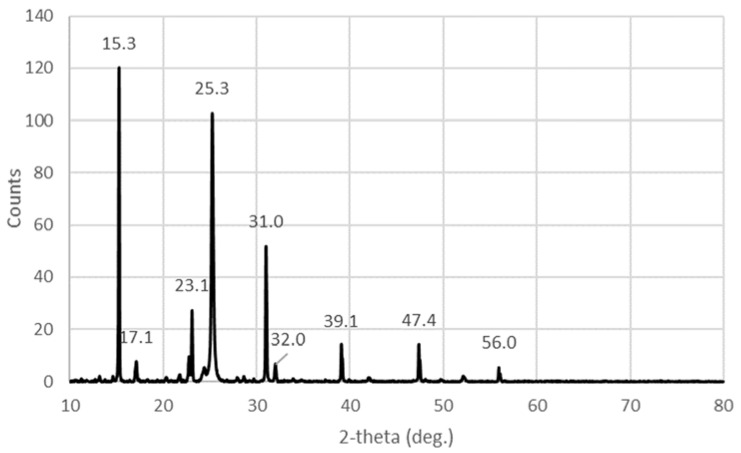
The XRD analysis of [Cu(12-crown-4)_2_]I_5_.

**Figure 5 molecules-27-06437-f005:**
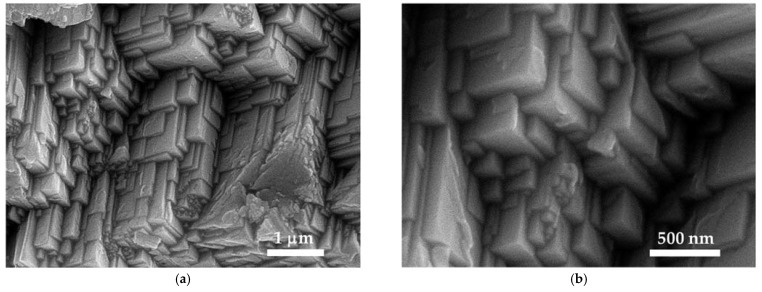
The scanning electron microscopy (SEM) of [Cu(12-crown-4)_2_]I_5_ (**a**) 1μm; (**b**) 500 nm.

**Figure 6 molecules-27-06437-f006:**
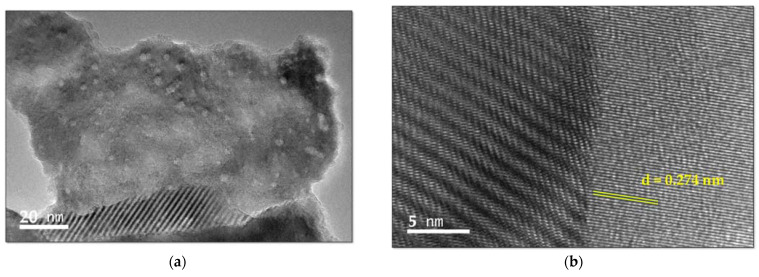
Transmission electron microscopy (TEM) of [Cu(12-crown-4)_2_]I_5_: (**a**) 20 nm; (**b**) 5 nm with average d-spacing of lattice fringes; (**c**) direction of the main lattice fringes in comparison to the black-white colored barrel-like structures; (**d**) Energy dispersive spectroscopy (EDS); layered images: (**e**) all atoms; and (**f**) I-atom.

**Figure 7 molecules-27-06437-f007:**
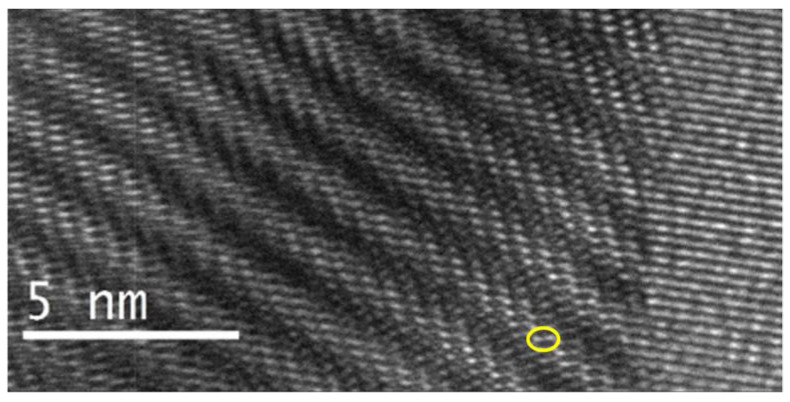
The transmission electron microscopy (TEM) of [Cu(12-crown-4)_2_]I_5_: A deeper look into the structure.

**Figure 8 molecules-27-06437-f008:**
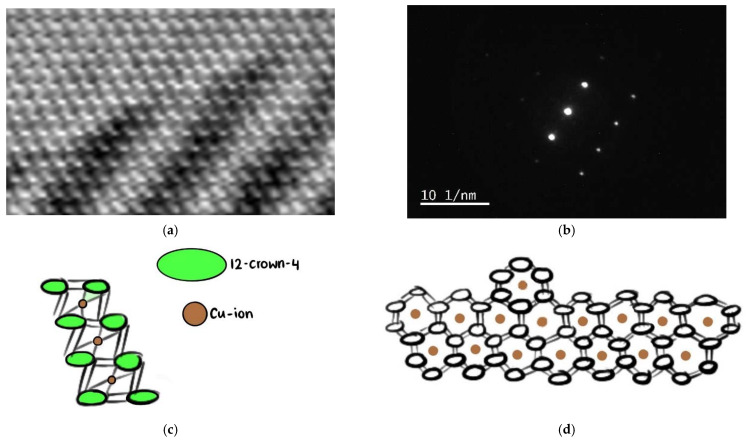
The transmission electron microscopy (TEM) of [Cu(12-crown-4)_2_]I_5_: (**a**) 1274% magnification of Figure 6c; (**b**) electron diffraction pattern along lattice fringe; and illustration of structures (**c**) ribbon; (**d**) honeycomb.

**Figure 9 molecules-27-06437-f009:**
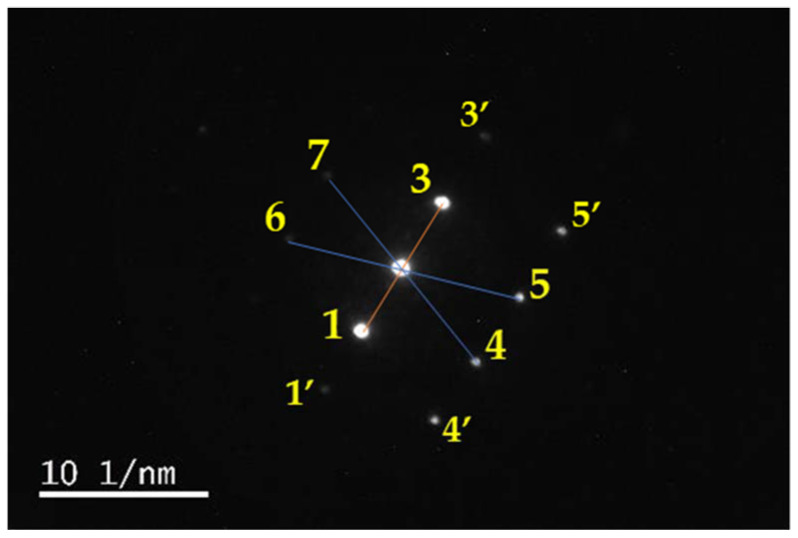
The transmission electron microscopy (TEM) of [Cu(12-crown-4)_2_]I_5_: Electron diffraction pattern along lattice fringes. Main central spots from down-left to up-right 1-2-3 (orange).

**Figure 10 molecules-27-06437-f010:**
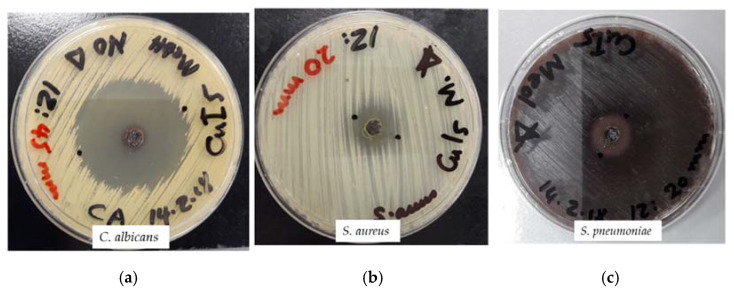
The Agar Well (AW) analysis of the title compound (with concentrations of 11, 5.5, and 2.75 µg/mL). From left to right: (**a**) *C. albicans* WDCM 00054; (**b**) *S. aureus* ATCC 25932; and (**c**) *S. pneumoniae* ATCC 49619.

**Figure 11 molecules-27-06437-f011:**
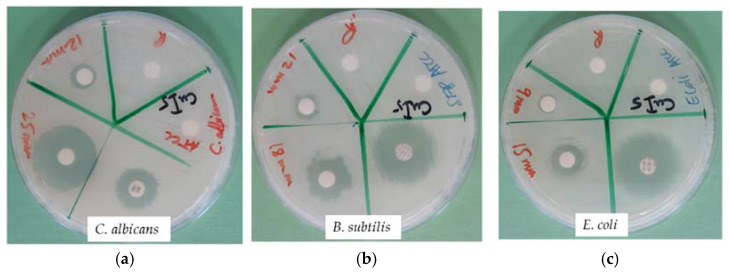
The disc diffusion (DD) studies with positive controls (antibiotic). From left to right: Title compound against (**a**) *C. albicans* WDCM 00054; (**b**) *B. subtilis* WDCM 00003; and (**c**) *E. coli* WDCM 00013.

**Table 1 molecules-27-06437-t001:** The UV-vis absorption signals in the samples [Cu(12-crown-4)_2_]I_5_ (1), [Rb(12-crown-4)_2_]I_5_ (2) [49], and previous reports (nm).

Group	1	2 [(I^−^)2(I_2_)]	[18]	[19]	[23]	[22]^exp^	[22]^calc^
I_2_/12-crown-4	204 vs		204	204			204
I_3_^−^/12-crown-4 I_3_^−^ [I_3_^−^^…^I_2_]	228 s,br 290 m,br 360 s,br	290 359	228 290 359	229 291 359	290 360	223 292 353	217 292 365/339Y
I^−^	201 vs	201	201				
I_5_^−^	445 vw,br		455	450	460		

vw = very weak, br = broad, s = strong, vs = very strong, m = intermediate, exp = experimental, and calc = calculated.

**Table 2 molecules-27-06437-t002:** The Raman shifts of iodine moieties in [Cu(12-crown-4)_2_]I_5_ (1), [Cu(H_2_O)_6_(12-crown-4)_5_]I_6_ x 2I_2_ (2) [50] and previous reports (cm^−1^) [22,23,28,39,42,45,47].

Group	1.	2	[42]	[45]	[39]	[47]	[22]	[23]	[28]
I_5_^−^ Type	V	Y	V	V	V	L	Linear	Linear	
Structure	[(I_3_^−^)(I_2_)]	[(I_3_^−^)2(I_2_)_0.5_]	[(I^−^)2(I_2_)]	[(I_3_^−^)(I_2_)]	[(I_3_^−^)(I_2_)]	[(I_3_^−^)(I_2_)]	[(I_5_^−^)]_n_	[(I_5_^−^)]_n_	
I_2_ as [I_2_^….^I^−^] or [I_2_^….^I_3_^−^]	s 170 ν_s_	vs 172 ν_s_	s 170 ν_s_ w 165^L^ ν_as_	177 ν_s_	178 ν_s_ 163 ν_as_	m 164 ν_s_	160 ν_as_	163 ν_as_	
I_3_^-^	m, sh [I-I-I^−^] 71 ν_2bend_ vs [I-I-I^−^] 112 ν_s_ w, sh [I-I^….^I^−^] 222 ν_as_		71 ν_2bend_ vs 111 ν_s_ w 222 ν_as_	67 ν_2bend_ 114 ν_s_	108 ν_s_ 211 ν_as_ 423	80 ν_2bend_ vs 111 ν_s_ s 127 ν_as_	70 ν_2bend_ 110 ν_s_ 220 ν_as_	108 ν_s_ 218 ν_as_	110 ν_s_ 217 ν_as_
I_5_^-^	m,sh 144 ν_as_	m,sh 142 ν_as_	w 144 ν_as_	151 ν_as_					147 ν_as_
as [I_2_^….^I_3_^−^]	vw, sh 334 ν_as_		334 ν_as_		334 ν_as_		318 ν_as_	337 ν_as_	331 ν_as_
12-crown-4	vw 2926	vw 2855							

ν = vibrational stretching, s = symmetric, as = asymmetric, s = strong, m = intermediate, sh = shoulder, v = very, and w = weak, red colored shifts belong to iodine in [**I_2_**^….^I_3_^−^].

**Table 3 molecules-27-06437-t003:** Comparative study with [Cu(12-crown-4)_2_]I_5_, [Cu(H_2_O)_6_(12-crown-4)_5_]I_6_ x 2I_2_ (2) [50], and previous investigations [45,47,49] (cm^−1^).

Compound	Type	[I_2_] (Å)	[I_3_^−^] (Å)	[I_2_^….^I_3_^−^] (Å)	[I_3_^−^] Angle (°)	[I_2_] (cm^−1^)	[I_3_^−^] (cm^−1^)	[I_5_^−^]/[I_2_^….^I_3_^−^] (cm^−1^)
[47] Polymeric Chain [(I_3_^−^)(I_2_)]_n_	L	2.75	2.83 3.05 2.94 *	3.38 3.40 3.39 *	177.45	m 164 ν_s_	80 ν_2bend_ vs 111 ν_s_ s 127 ν_as_	
[45] Polymeric [(I_3_^−^)(I_2_)]_n_	V	2.74	2.90 2.93 2.92 *	3.41	176.43	m 177 ν_s_	67 ν_2bend_ vs 114 ν_s_	vs 151 ν_as_
[Cu(H_2_O)_6_(12-crown-4)_5_]I_6_ x 2I_2_ Polymeric chain	Y	2.76 2.77	2.80 3.06	3.38 3.38	176.46	vs 172 ν_s_		m,sh 142 ν_as_
[(I_3_^−^)2(I_2_)_0.5_]_n_		2.77*	2.93*	3.38*				
[Rb(12-crown-4)_2_]I_5_ Isolated [(I^−^)2(I_2_)]	V	2.79	2.82 3.09 2.96 *	3.09	177.30	m 163 ν_s_ w,sh 180 ν_s_	72 ν_2bend_ m,br 96 ν_s_ w,br 190-230 ν_as_	s,br 142 ν_as_
[Cu(12-crown-4)_2_]I_5_ Chain	V	≥2.77 ≤2.75	≥2.92 ≤2.93	≥3.39 ≤3.41	~180 ≥176.43 ^+^ ≤177.45 ^+^	s 170 ν_s_	m,sh [I-I-I^−^] 71 ν_2bend_ vs [I-I-I^−^] 112 ν_s_ w,sh [I-I^….^I^−^] 222 ν_as_	m,sh 144 ν_as_ vw,sh 334 ν_as_

* average. ^+^ expected bond angles for asymmetric triiodide units. ν = vibrational stretching, s = symmetric, as = asymmetric, s = strong, m = intermediate, sh = shoulder, v = very, w = weak, and br = broad. Red/Blue shift in comparison to other bands in Table 3.

**Table 4 molecules-27-06437-t004:** The FTIR analysis [cm^–1^] of 12-crown-4 (1) [50], [Cu(12-crown-4)_2_]I_5_ (2), [Cu(12-crown-4)_5_(H_2_O)_6_]I_6_ x 2I_2_ (3) [50], and [Rb(12-crown-4)_2_]I_5_ (4) [49].

	ν_1_ (O–H)*_s_ ν_2_ (O–H)*_as_	ν (C–H)_as_	ν (CH_2_)_as,s_	ν (C–H)_s_ ν (O–H)*	δ (C–H)_as_ ν_3_ (O–H)*	δ (C–C)	δ (C–H) δ (C–C)	ν (C–O) ν (C–O-C)	ν (CH–CH) ν (O–H)*
1		w 2950	w 2900	w 2850	w 1450	w 1365	w 1290	w 1250 vs 1125 s 1100 m 1075 m 1025 w 975 m 919	m 848 vw 815
2	w,sh 3393 * w 3240 * w 3179 *	w 2953	w 2905	w 2859	w 1445 vw 1638 *	w 1362	w 1287	w 1248 m 1124 vs 1090 s 1069 s 1022 m 916	s 841 m 810 ν_56_ m 588 m 548 ν_57_/ν_16_ m 521
3	3746 *_s_ 3418 *_as_	2951	2905	2863 2777 * 2733 *	1443 1642 *	vw 1360	1287	1243 1133 1092 m 1022 911	m 844 549 628 *
4		2954	2906	2865	1444	1363	1289	1245 1134 1095 1023 915	849

* Signals related to vibrational modes of hydroxyl groups due to hydrogen bonding/intermolecular bonded alcohol and hydration. ν = vibrational stretching, δ = deformation, _s_ = symmetric, _as_ = asymmetric, vs = very strong, s = strong, m = intermediate, w = weak, vw = very weak, and sh = shoulder.

**Table 5 molecules-27-06437-t005:** The XRD analysis of the samples [Cu(12-crown-4)_2_]I_5_ (1), pure I2 [25], and previous reports (2Theta^o^) [30,33].

Group	1	[25]	[33]	[29]	[30]
I_2_	23.1 w 25.3 s 31 m 39.1 vw 47.4 vw	24.5 s 25 s 28 vs 37 w 38 w 43 w 46 m	22.8 m 23.8 m 25.5 m 27 m 30 m 37 vw 46 w	25 29 36	
I_5_^−^	15.3 vs				8 m 10 m 13.4 vs

w = weak, s = strong, m = intermediate and v = very.

**Table 6 molecules-27-06437-t006:** The antimicrobial testing of antibiotics (A), title compound (B), and [Cu(12-crown-4)_5_(H_2_O)_6_]I_6_ x 2I_2_ (C) [50] by Agar Well (AW) and disc dilution studies (1,2,3). The ZOI (mm) against microbial strains was determined by diffusion assay.

Strain	Antibiotic	A	AW^B^	AW^C^	1^+B^	2^+B^	3^+B^	1^+C^	2^+C^	3^+C^
*S. pneumoniae* ATCC 49619	G	18	20	20	25	0	0	19	0	0
*S. aureus* ATCC 25923	G	28	20	23	11	0	0	35	14	0
*S. pyogenes* ATCC 19615	C	25	25	20	13	0	0	21	0	0
*E. faecalis* ATCC 29212	CTX	25	20	19	16	0	0	18	0	0
*B. subtilis* WDCM 00003	S	21	19	21	18	12	0	33	11	0
*P. mirabilis* ATCC 29906	G	30	0	0	20	0	0	15	0	0
*P. aeruginosa* WDCM 00026	CTX	23	17	16	12	0	0	12	0	0
*E. coli* WDCM 00013	A	23	15	15	15	9	0	25	8	0
*K. pneumoniae* WDCM 00097	CTX	30	NA	NA	16	8	0	24	7	0
*C. albicans* WDCM 00054	NY	16	45	51	25	12	0	53	14	0

Agar well (AW) diffusion studies (20 mg crystals of title compound in a 6 mm diameter well) and disc diffusion studies (6 mm disc impregnated with 2 mL of 50 µg/mL (1), 2 mL of 25 µg/mL (2), and 2 mL of 12.5 µg/mL (3)). A Amikacin (30 µg/disc), G Gentamicin (30 µg/disc), CTX (Cefotaxime) (30 µg/disc), NY (Nystatin) (100 IU), C Chloramphenicol (10 µg/disc), and Streptomycin (10 µg/disc). Grey shaded area represents Gram-negative bacteria. 0 = Resistant. No statistically significant differences (*p* > 0.05) between row-based values through Pearson correlation.

## Data Availability

Not applicable.

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
