# Peer review of "Antimicrobial V-Shaped Copper(II) Pentaiodide: Insights to Bonding Pattern and Susceptibility"

_molecules, 2022, doi:10.3390/molecules27196437_

Round 1

Reviewer 1 Report

Edis and co-worker synthesized the novel pentaiodides and evaluated for its antibacterial potential. The [Cu(12-crown-4)2]I5 was characterized carefully using UV-Vis Spectroscopy, Raman Spectroscopy, Fourier-Transform Infrared (FTIR) Spectroscopy, X-ray Diffraction (XRD), Scanning Electron Microscope (SEM), Transmission electron microscopy (TEM). The overall presentation of the work is good. Few minor points need to be take care.

1.      Abstract needs to be concise

2.     Reference should be given for the statement “Our last investigation resulted in an interesting Y-shaped polymeric pentaiodide [Cu(H2O)6(12-crown-4)5]I6 x 2I2 with very good antimicrobial activities against 10 selected pathogens” If published(line no. 85)

3.     In line no.132, authors should use single crystal X-Ray diffraction instead of X-Ray methods as they have done the powder XRD.

4.     Improve the resolution of Figure 1,2 and 4

5.     Line no. 112, model for drug interaction is repeating again and again… should modify the statement

6.     The representation of 12-crown 4 should be same throughout the paper (line 125, 126)

7.     Does the authors have conducted the stability study of the prepared complex? if not they should provide some relevant data regarding the stability of the compound under study.

Author Response

 Dear Reviewer, thank you so much for your kind and honest efforts to improve our manuscript. We really deeply appreciate your valuable input and hope, we fulfilled your expectations.

Comments and Suggestions for Authors

Edis and co-worker synthesized the novel pentaiodides and evaluated for its antibacterial potential. The [Cu(12-crown-4)2]I5 was characterized carefully using UV-Vis Spectroscopy, Raman Spectroscopy, Fourier-Transform Infrared (FTIR) Spectroscopy, X-ray Diffraction (XRD), Scanning Electron Microscope (SEM), Transmission electron microscopy (TEM). The overall presentation of the work is good. Few minor points need to be take care.

  1. Abstract needs to be concise. Dear Reviewer, thanks for your comment. We kept part, which we felt are impervious, removed one sentence in the end and added instead

“The single crystal structure is highly arranged across the lattice fringes in form of ribbons or honeycombs.” This was missing related to the structure.

  1. Reference should be given for the statement “Our last investigation resulted in an interesting Y-shaped polymeric pentaiodide [Cu(H2O)6(12-crown-4)5]I6 x 2I2 with very good antimicrobial activities against 10 selected pathogens” If published(line no. 85). Thank you. Done. It is [50].
  2. In line no.132, authors should use single crystal X-Ray diffraction instead of X-Ray methods as they have done the powder XRD. Done. Thanks
  3. Improve the resolution of Figure 1,2 and 4. Dear Reviewer, we tried our best to do so and hope you are satisfied, but it seems, it is still looking the same. We hope, you accept our apologies for not getting higher resolution.
  4. Line no. 112, model for drug interaction is repeating again and again… should modify the statement. Dear Reviewer, you are right. We removed the sentence, because it was anyhow a repetition of the same fact elaborated in the introduction. Thank you so much.
  5. The representation of 12-crown 4 should be same throughout the paper (line 125, 126). Dear Reviewer, we went through the manuscript and checked, keeping it 12-crown-4, instead of 12-Crown-4, which was available in the references section as [49]. Thanks again !
  6. Does the authors have conducted the stability study of the prepared complex? if not they should provide some relevant data regarding the stability of the compound under study. Dear Reviewer, we conducted stability study for 18 months in closed glas bottle at 7 degrees. Thank you so much for giving us the chance to mention this in the manuscript. We added this information at section 3.3 Characterization of [Cu(12-crown-4)2]I5 as sentence “The complex was stored in the fridge for 18 months in closed glas bottle without changing its composition and morphology.”

Best regards

Thank you so much again indeed

Zehra Edis

Reviewer 2 Report

The manuscript submitted by Edis and Haj Bloukh is not suitable for publication. 

The introduction lacks structure. UV-vis assignation is incomplete; the assignation proposed in table 1 does not correspond with the literature (see supplementary material of https://doi.org/10.1039/C7EE00954B). The authors should include UV spectra of I2, I-, I3-, and I5 in the same experimental conditions for a specific assignation. The units used in Table 2 for Raman vibration assignation do not match those used in the main text.  

The structural description from TEM, STEM, and EDS considers copper ion and 12-crow-4 but not pentaiodide; why? The authors should explain.

On the other hand, the authors start with CuI, 

The binding constant for Cu(II) and oxygen atoms from 12-crow-4 are low and are even lower for Cu(I). Therefore, which is the oxidation state proposed for the copper ion in the title compound? which is the interaction mode proposed for the copper ion and 12-crown-4? Which is the proposed interaction between copper ion ad iodine species?

The binding constant for Cu(I) and iodine species are higher than those with Cu(II). 

The oxidation state of the copper ion will also modify the antibacterial activity, therefore, is mandatory to show the antibacterial activity of the copper ion in the corresponding oxidation state.    

The authors conclude that a completely different morphology and topology were found with slightly different reaction conditions but did not provide spectroscopical evidence of no hydration of the new compounds nor the stability of the compound in solution. The UV spectroscopy shows triiodide predominantly.

The authors attribute the antibacterial activity to Iodine, triiodide, and iodide release. Still, they do not show the biological activity of each of these compounds individually or the components of the title compound to support their proposal by direct comparison with the antimicrobial activity of the title compound. 

It is essential to mention that the suggested comparison neither exists in the previous reports of this research group. Also, the same inconsistencies in the characterization were found. 

I believe there is no new significant antimicrobial result compared with the previous report. Without the antimicrobial effects, the authors could modify the manuscript and report the structural characterization considering the last comments submitting the new report to Inorganics or another journal that fits its aim and scope. 

Author Response

Dear Reviewer,

thank you so much for your comments. We improved our manuscript and added needed information due to your valuable comments.

Best regards

Zehra
